# An Innovative Correction–Fusion Approach for Multi-Satellite Precipitation Products Conditioned by Gauge Background Fields over the Lancang River Basin

Linjiang Nan [1,2], Mingxiang Yang [2,*], Hao Wang [2], Hejia Wang [2] and Ningpeng Dong [2]

1  The College of Water Resource and Hydropower, Sichuan University, Chengdu 610065, China; nanlinjiang917@163.com
2  Department of Water Resources, China Institute of Water Resources and Hydropower Research, Beijing 100038, China; wanghao@iwhr.com (H.W.); hjwang@iwhr.com (H.W.); dongnp@iwhr.com (N.D.)
*  Correspondence: yangmx@iwhr.com; Tel.: +86-18046555306

**Abstract:** Satellite precipitation products can help improve precipitation estimates where ground-based observations are lacking; however, their relative accuracy and applicability in data-scarce areas remain unclear. Here, we evaluated the accuracy of different satellite precipitation datasets for the Lancang River Basin, Western China, including the Tropical Rainfall Measuring Mission (TRMM) 3B42RT, the Global Precipitation Measurement Integrated Multi-satellitE Retrievals (GPM IMERG), and Fengyun 2G (FY-2G) datasets. The results showed that GPM IMERG and FY-2G are superior to TRMM 3B42RT for meeting local research needs. A subsequent bias correction on these two datasets significantly increased the correlation coefficient and probability of detection of the products and reduced error indices such as the root mean square error and mean absolute error. To further improve data quality, we proposed a novel correction–fusion method based on window sliding data correction and Bayesian data fusion. Specifically, the corrected FY-2G dataset was merged with GPM IMERG Early, Late, and Final Runs. The resulting FY-Early, FY-Late, and FY-Final fusion datasets showed high correlation coefficients, strong detection performances, and few observation errors, thereby effectively extending local precipitation data sources. The results of this study provide a scientific basis for the rational use of satellite precipitation products in data-scarce areas, as well as reliable data support for precipitation forecasting and water resource management in the Lancang River Basin.

**Keywords:** Fengyun 2G; TRMM 3B42RT; GPM IMERG; suitability assessment; deviation correction; data fusion

## 1. Introduction

Precipitation is the most important driving factor of the terrestrial hydrological cycle and a primary contributor to uncertainties in the prediction of hydrological fluxes and states [1]. Accurate precipitation observations and estimates are therefore crucial for hydrograph prediction, flood forecasting, drought monitoring, and water resource allocation and management. Areas such as the Lancang River Basin in Western China, however, lack the ability to measure and predict precipitation and therefore struggle to understand local hydrological characteristics. Moreover, advances in hydropower development in the basin and the worsening impact of climate change have increased the demand for high-quality precipitation data [2].

Precipitation is one of the most difficult atmospheric variables to measure due to its large spatiotemporal variability and non-normal distribution. Conventionally, ground gauges are the primary means of obtaining precipitation data. However, owing to financial or topographical limitations, ground-based precipitation observations can be infrequent or nonexistent, particularly in developing countries and remote regions. Following recent

rapid developments, satellite precipitation products (SPPs) have exhibited an unprecedented ability to provide spatiotemporally continuous estimates of precipitation, with advantages such as a broad coverage, high temporal and spatial resolution, and no terrain or climatic limitations [3]. As a result, recent research has focused on the applicability of mainstream SPPs, including the Tropical Rainfall Measuring Mission (TRMM) and Global Precipitation Measurement (GPM). Previous evaluations of the TRMM have been diverse in scope, from near-global [4] to smaller-scale analyses of river basins or regions, including the Philippines [5], Far East Asia [6], East Sichuan [7], Hanjiang River Basin [8], and Poyang Lake Basin [9] in China. These studies have the following findings in common: (1) The TRMM performs much better on a monthly than on a daily scale; (2) the accuracy of TRMM satellite precipitation data is poor in winter but high in spring, summer, and autumn; (3) and near-real-time products (e.g., TRMM 3B42V6 and 3B42V7) have a lower accuracy than post-processing products (e.g., TRMM 3B42RT) [10,11]. Launched in 2014, the GPM Integrated Multi-satellitE Retrievals (IMERG) product is the next generation of global SPPs after the TRMM. Previous research on GPM IMERG has focused on performance comparisons with the previous generation of TRMM series products. While several studies [12,13] have demonstrated a substantial improvement in accuracy when transitioning from the TRMM to GPM, others [14] have found no improvement or even a decline in accuracy compared to the TRMM. Moreover, no previous studies have compared the accuracy of the Early Run, Late Run, and Final Run estimates in GPM IMERG or determined their applicability to data-poor areas in Western China, such as the Lancang River Basin.

Satellite precipitation inversion is an indirect estimation method limited by several inherent errors [15]. Further data correction based on the spatiotemporal distribution characteristics and error structure of satellite precipitation data is required to not only improve data accuracy, but also to optimize next-generation product data sources, update and iterate algorithms, and provide strong data support for precipitation prediction in data-poor areas such as the Lancang River Basin. Common bias correction methods include the stepwise regression, geographically weighted regression [16], error decomposition [17], the statistical bias correction method [18], the bidirectional long short-term memory cycle correction model [19], the scale method [20], the combined linear scaling and quantile-mapping cumulative distribution function [21], the empirical cumulative distribution [22], the linear regression [23], and multi-technology coupling [24,25]. These correction methods can be classified as either local corrections or machine learning methods. A local correction is simple to perform; however, it does not consider the spatial heterogeneity of the data, resulting in large errors. Conversely, machine learning methods require extensive ground observation data, are highly dependent on data quality, and are limited by several difficulties and in areas lacking data [26]. Furthermore, although the above correction methods improve the data quality, the potential for further data fusion requires further investigation.

Lancang River Basin has a steep and complex topography, so the number of rainfall and meteorological stations in this region is very limited, and their spatial distribution is extremely uneven. At the same time, the precipitation observation period is limited, and it is difficult to reproduce the spatial and temporal distribution of precipitation completely. Remote sensing precipitation data cover a large range, are basically not limited by terrain or meteorological and climatic conditions, and can fully describe the spatiotemporal distribution of precipitation. Therefore, remote sensing technology, including the evaluation of satellite precipitation products, correction, and fusion, has a high research value in Lancang River Basin. It can be used to make up for the shortage of meteorological rainfall stations and obtain relatively complete remote sensing precipitation data with long-term series and a large spatial coverage. It is of great practical significance and social value to water resource planning and management, drought and flood forecasting, hydrological simulations, and hydropower development and utilization in Lancang River Basin.

Therefore, in this study, we propose a correction–fusion method for SPP data that considers the unique geographical environment and data conditions of the Lancang River Basin. This method involves a novel window sliding correction method that considers the

spatial distribution of data. After evaluating the original datasets derived from three SPPs, we perform local to global corrections based on ground measurements of precipitation. Data fusion is then again performed to further improve the precision and reliability of the SPP datasets. This research has high practical significance for precipitation forecasting and water resource management in data-poor areas.

## 2. Study Area and Data

### 2.1. Study Area

The Lancang River Basin (Figure 1) is located in Southwest China (longitude 93°48′–101°51′ east and latitude 21°06′–33°48′ north). The upper and lower reaches are wide, whereas the middle reaches are narrow and slender, with a ribbon distribution from the north to south. The terrain is high in the northwest and low in the southeast, with a high level of variability. The upper reaches lie in Qamdo, Tibet, in the Tanggula fold belt of the Qinghai–Tibet Plateau, at an elevation of more than 4500 m, which is characterized by a relatively preserved plateau landform. The middle reaches from Qamdu to Sijia village in Yunnan Province belongs to the high mountain valley area, with a valley floor elevation between 1230 m and 2200 m and a typical relative elevation difference of approximately 2000 m. Sijia village lies in the lower reaches of the basin, with the lowest riverbed elevation being 486 m [27].

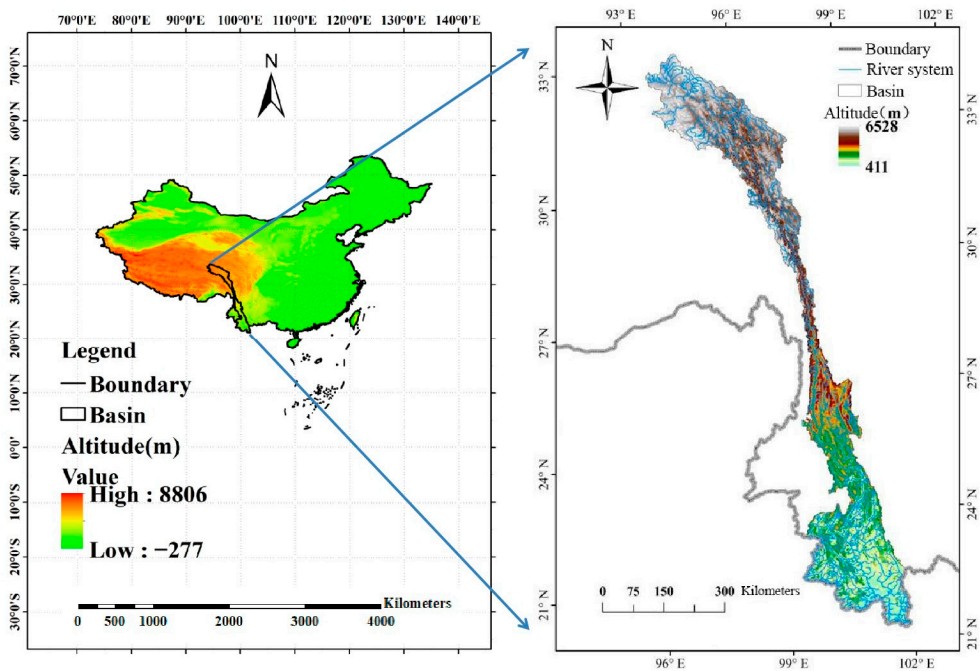

**Figure 1.** Topographic map of the study area.

The Lancang River Basin has a plateau mountain climate and a subtropical monsoon climate, with a rainy season from May to October and a dry season from November to April. The upper reaches of the Qinghai–Tibet Plateau are characterized by low temperatures and minimal rain. The middle reaches represent a transitional climate zone from cold to subtropical, where the vertical climate characteristics are significantly altered by high levels of precipitation. The lower reaches have high temperatures and abundant precipitation owing to a subtropical climate [28].

### 2.2. Data

#### 2.2.1. Satellite Datasets

GPM IMERG is the next-generation product of the TRMM, and its performance relative to that of the TRMM has been widely studied. According to previous studies, we also

analyze the performance of a Chinese domestic product, FY-2G SPP. Information on all SPPs included in this study is shown in Table 1.

**Table 1.** Basic information of the five satellite precipitation products analyzed in this study.

| Name | Spatial Resolution | Temporal Resolution | Period |
|---|---|---|---|
| TRMM 3B42RT | $0.25° \times 0.25°$ | daily | 2016–2019 |
| GPM IMERG Early Run | $0.1° \times 0.1°$ | daily | 2016–2020 |
| GPM IMERG Late Run | $0.1° \times 0.1°$ | daily | 2016–2020 |
| GPM IMERG Final Run | $0.1° \times 0.1°$ | daily | 2016–2020 |
| FY 2G | $0.1° \times 0.1°$ | daily | 2016–2020 |

The TRMM satellite was jointly developed by the United States and Japan and launched in 1997. Precipitation data retrieved by the TRMM satellite have the advantages of a wide coverage, high spatial resolution, and favorable consistency; thus, this dataset is widely used in scientific research. In this study, we used TRMM 3B42RT SPPs from 2016 to 2019, which have a spatial resolution of $0.25° \times 0.25°$ and a temporal resolution of one day.

To produce more accurate satellite precipitation estimates at finer spatiotemporal resolutions, the National Aeronautics and Space Administration and Japan Aerospace Exploration Agency jointly launched the GPM mission on 27 February 2014, providing a new generation of precipitation products based on the TRMM. Despite inheriting the mature algorithm and detection technology of the TRMM, GPM exhibits an improved monitoring performance, which not only improves the spatial and temporal resolution but also generates precipitation data in a larger spatial range [29]. This study adopted the daily-scale data of GPM IMERG V06 from 2016 to 2020, which have a spatial resolution of $0.1° \times 0.1°$ and include Early, Late, and Final Runs. The GPM IMERG Early Run (Early) uses only the forward morphing algorithm to invert precipitation and can provide fast estimates 4 h after observation. Subsequently, with an increase in observed data, the GPM IMERG Late Run (Late) is obtained by combining both forward and backward morphing algorithms, with a minimum data latency of approximately 12 h. Finally, monthly-scale ground observation data were used for calibration to obtain the GPM IMERG Final Run (Final), with a data delay of approximately 3.5 months for the 30 min dataset. Both TRMM 3B42RT and GPM IMERG SPPs, derived from https://disc.gsfc.nasa.gov/ (accessed on 4 January 2021), are subject to strict quality control monitoring.

The FY-2G satellite was successfully launched from the Xichang Satellite Launch Center in China on 31 December 2014 and was fixed over the equator at 99.5°E on 6 January 2015. On 1 June 2015, it drifted to 105°E longitude, replacing the E satellite that had been in service for an extended period, thus becoming the main operational satellite and strengthening China's meteorological monitoring capability during the flood season. FY-2G satellite rainfall data from 2016 to 2020, which have a spatial resolution of $0.1° \times 0.1°$, were obtained from http://satellite.nsmc.org.cn/ (accessed on 8 March 2021) and subject to strict quality control monitoring.

### 2.2.2. Rain Gauge Data

Daily rain gauge precipitation data from a total of 176 observation stations, which underwent strict quality control, were provided by an automatic system built by the Lancang River Group Company and the China Meteorological Data Network (http://data.cma.cn/ (accessed on 7 September 2021)). To ensure abundant and complete data from the ground stations and consider the time series of various SPPs, we selected a dataset of almost five years from 1 January 2016 to 31 December 2020; each observation station had a complete data time series during this period. To match the temporal characteristics of the measured ground station data with those of the satellite precipitation data, the daily observation period of the measured precipitation data was set to Beijing time (UTC+8), that is, from 08:00 on the same day to 08:00 on the next day.

## 3. Methods

### 3.1. Evaluation Indexes

In this study, statistical and classification indices were used to evaluate the accuracy of various SPPs relative to that of ground-measured data. For the statistical evaluation, the correlation coefficient (CC), root mean square error (RMSE), and relative bias (RB) were used to evaluate the ability of the SPP datasets to capture precipitation characteristics over time. For the classification evaluation, the detection probability (POD), false alarm rate (FAR), and critical success index (CSI) were used to evaluate the ability of the SPP datasets to detect precipitation events. Previous studies considered 1 mm/day as the threshold for distinguishing days with and without rain [30]. To reflect the ability of SPPs to detect precipitation events in more detail, four precipitation thresholds in this study were used as follows: 0.1, 10, 25, and 50 mm/d, which were used to delimit rain conditions into "rain/no rain", "light rain", "moderate rain", "heavy rain", and "torrential rain" [31]. The formulas, ranges, and optimal values of the evaluation indicators are listed in Table 2.

**Table 2.** Description of evaluation indexes used in this study.

| Statistics | Formula | Range | Optimal Value |
|:---:|:---:|:---:|:---:|
| CC [32] | $CC = \dfrac{\sum \left(X_i - \overline{X}\right)\left(Y_i - \overline{Y}\right)}{\sqrt{\sum \left(X_i - \overline{X}\right)^2 \left(Y_i - \overline{Y}\right)^2}}$ | $[-1, 1]$ | 1 |
| RMSE [33] | $RMSE = \sqrt{\dfrac{1}{n}\sum\limits_{i=1}^{n}\left(X_i - Y_i\right)^2}$ | $[0, +\infty]$ | 0 |
| RB [34] | $RB = \dfrac{\sum_{i=1}^{n}(Y_i - X_i)}{\sum_{i=1}^{n} X_i} \times 100$ | $(-\infty, +\infty)$ | 0 |
| POD [35] | $POD = \dfrac{H}{H+M}$ | $[0, 1]$ | 1 |
| FAR [36] | $FAR = \dfrac{F}{H+F}$ | $[0, 1]$ | 0 |
| CSI [37] | $CSI = \dfrac{H}{H+M+F}$ | $[0, 1]$ | 1 |
| ETS [38] | $H_s = \dfrac{(H+M)(H+F)}{H+M+F+Z}$ <br> $ETS = \dfrac{H-H_s}{H+M+F-H_s}$ | $[-\frac{1}{3}, 1]$ | 1 |
| FBI [38] | $FBI = \dfrac{H+F}{H+M}$ | $[0, +\infty]$ | 1 |

Notation: $n$ indicates the number of data pairs used in the accuracy evaluation. $X_i$ represents the observed precipitation value of ground station $i$; $Y_i$ represents the pixel value of the satellite precipitation data grid where the ground station is located. $H$ represents the number of precipitation events successfully captured by ground observation stations and satellites at the same time under a specific threshold value. $M$ represents the number of precipitation events successfully captured by ground observation stations and unsuccessfully captured by satellites under a specific threshold value. $F$ represents the number of precipitation events captured by satellites but not observed by ground stations at a specific threshold.

### 3.2. Merging Technique

The proposed correction–fusion method included two parts, window sliding data correction and Bayesian data fusion, and was employed to improve data quality through secondary correction, thereby providing more reliable data support for subsequent local precipitation forecasts.

#### 3.2.1. Window Sliding Data Correction

Current correction methods for satellite precipitation data mainly include local and global corrections. The main global correction methods include linear regression and average deviation correction [39], and the main local correction methods include geographically weighted regression, Bayesian correction, and ordinary cooperative kriging. Common satellite precipitation data correction techniques usually do not consider the temporal and spatial variability of precipitation, which leads to uncertainty in the results.

Based on the above techniques, we proposed a novel window sliding data correction method that considers the spatial distribution of precipitation data and is based on the

structural characteristics of the SPPs and ground-measured data. Ground scatter data were included in the satellite grid in space for a certain time. Tesfagiorgis et al. [40] argued that a simple way to reduce the error of one precipitation product relative to another reference product is to multiply the rainfall of the first product by a "deviation factor" to optimize the corresponding relationship whereby the two products overlap. Therefore, we considered ground observation data as the reference value and used the relationship between the two types of data for calibration [41]. The main steps were as follows: (1) pre-process the satellite raster data and ground-measured precipitation data; (2) starting from the first grid of satellite raster data, fix the vertices and set the window width; (3) correct the satellite raster data in the initial window, then complete the error correction of all satellite raster data by sliding the initial window; (4) and increase the window width by one grid unit step by step, then repeat steps (2) and (3) for correction. The final correction effect corresponding to each window width was evaluated until the best correction matrix was selected and used as the correction dataset of SPPs. The main concepts and implementation process of the correction method are illustrated in Figure 2.

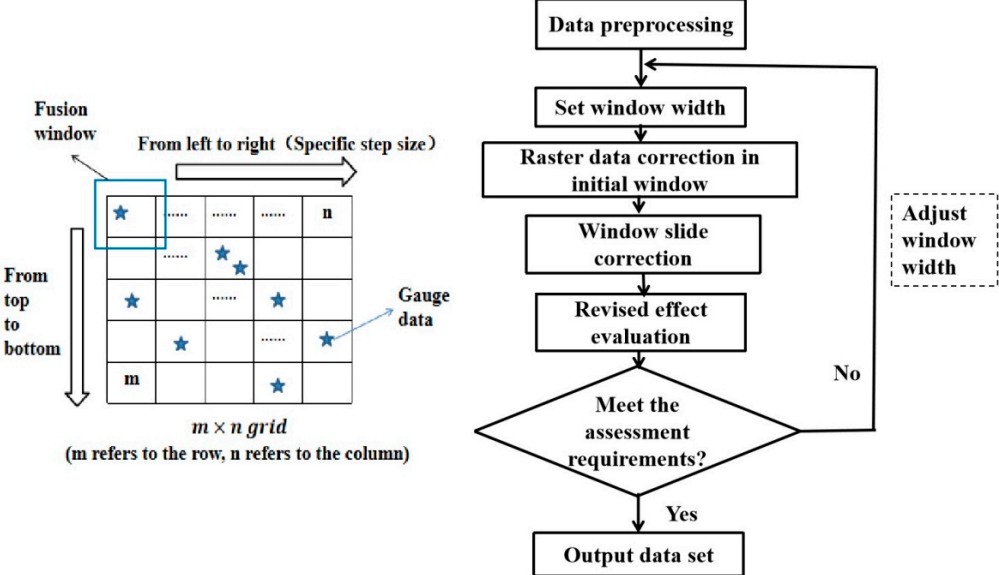

**Figure 2.** Flow chart of the proposed window sliding data correction method.

The proposed method calculated the arithmetic average of the raster data and ground-measured data in the correction window. The calculation formula was as follows:

$$\bar{x} = \frac{1}{n} \times (x_1 + x_2 + \ldots + x_n) \tag{1}$$

$$\bar{y} = \frac{1}{n} \times (y_1 + y_2 + \ldots + y_n) \tag{2}$$

where $x_1, x_2, \ldots, x_n$ are the measured data of ground stations in the window, and $y_1, y_2, \ldots, y_n$ are the raster data contained in the window. The relationship between raster data in the window and ground-measured data was expressed by the ratio $b$ according to the following calculation formula:

$$b = \frac{\bar{x}}{\bar{y}} \tag{3}$$

The correction result of the raster data in the window was obtained by multiplying the original raster data by the ratio $b$ according to the following calculation formula:

$$B_{m \times n} = b \times A_{m \times n} \tag{4}$$

where the original raster data in the window are denoted by $A_{m \times n} = [\ ]$, the revised raster data are denoted as $B_{m \times n}$, $m$ refers to the row, and $n$ refers to the column.



3.2.2. Bayesian Data Fusion

After performing SPP data correction, we performed data fusion via Bayesian model averaging (BMA), a statistical processing method based on Bayesian theory [42,43]. Assuming that $B$ is the data fusion result, $D$ is the measured rainfall data, and $f = [f_1, f_2, \ldots, f_k]$ represents the different SPP datasets, we calculated the BMA as follows:

$$p(B|D) = \sum_{k=1}^{K} p(f_k|D) \cdot p_k(B|f_k,\ D) \tag{5}$$

where $p(f_k|D)$ represents the posterior probability of the category $K$ SPP dataset after determining the measured data $D$, which reflects the matching degree between $f_k$ and measured precipitation. In fact, $p(f_k|D)$ is the weight, $\omega_k$, of BMA. The higher the SPP precision, the greater the optimal weight. In addition, the weights of all SPPs were positive, with a sum of one. Here, "$p_k(B|f_k, D)$" refers to the posterior distribution of fusion results under the given conditions of satellite precipitation data $f_k$ and measured precipitation data $D$.

This study utilized the expectation maximization (EM) algorithm to solve the parameters of the above Bayesian fusion model, which is an effective method for calculating the BMA based on the assumption that K-class datasets obey a normal distribution. Therefore, before using the EM algorithm, it is necessary to perform a normal conversion of ground-measured and satellite precipitation data. The calculation principle is as follows.

The logarithmic likelihood function of $\theta$ ($\theta = \{\omega_k, \sigma_k^2, k = 1,2,\ldots,K\}$) can be expressed as follows:

$$l(\theta) = \log(p(B|D)) = \log\left(\sum_{k=1}^{k} \omega_k \cdot g(B|f_k, \sigma_k^2)\right) \tag{6}$$

where $g(B|f_k, \sigma_k^2)$ represents the mean of $f_k$ and the variance of $\sigma_k^2$ in the normal distribution. The above formula cannot easily obtain the analytical solution of "$\theta$", whereas the EM algorithm can obtain the maximum likelihood value through the iterative process of expectation and maximization until convergence, thus obtaining "$\theta = \{\omega_k, \sigma_k^2, k = 1, 2, \ldots, K\}$". In the numerical solution of the EM algorithm, the hidden variable $z_k^t$ is used to assist in the BMA weight calculation. The detailed steps of the EM algorithm used to solve the BMA parameters are listed in Table 3.

**Table 3.** Description of expectation maximization algorithm steps used to solve the Bayesian model's averaging parameters.

| Step | Methods | Formula |
|------|---------|---------|
| 1 | Initialize ($Iter = 0$) | $\omega_k^{(0)} = 1/K,\ \sigma_k^{2(0)} = \frac{\sum_{k=1}^{k} \sum_{t=1}^{NT} (Y^t - f_k^t)^2}{K \cdot NT}$ |
| 2 | Calculate the initial likelihood value | $l(\theta)^{(0)} = \sum_{t=1}^{NT} log\left( \sum_{k=1}^{K} (\omega_k^{(0)} \cdot g(B|f_k^t \cdot \sigma_k^{2(0)})) \right)$ |
| 3 | Calculate hidden variables ($Iter = Iter + 1$) | $Z_k^{t(Iter)} = \frac{g\left(B\big|f_k^t, \sigma_k^{2(Iter-1)}\right)}{\sum_{k=1}^{K} g\left(B\big|f_k^t, \sigma_k^{2(Iter-1)}\right)}$ |
| 4 | Calculate the weight | $\omega_k^{(Iter)} = \frac{1}{NT}\left(\sum_{t=1}^{NT} Z_k^{t(Iter)}\right)$ |
| 5 | Calculate the error | $\sigma_k^{2(Iter)} = \frac{\sum_{t=1}^{NT} Z_k^{t(Iter)} \cdot (Y^t - f_k^t)^2}{\sum_{t=1}^{NT} Z_k^{t(Iter)}}$ |
| 6 | Calculate the likelihood value | $l(\theta)^{(Iter)} = \sum_{t=1}^{NT} log(\sum_{k=1}^{K} (\omega_k^{(Iter)} \cdot g(B|f_k^t, \sigma_k^{2(Iter)})))$ |
| 7 | Test the convergence | $if\ l(\theta)^{(Iter)} - l(\theta)^{(Iter-1)} < \varepsilon$ |

Notation: *Iter* is the number of iterations; *NT* is the rated data length; and $Y^t, f_k^t$ are the ground-observed precipitation and *k*-satellite precipitation datasets at time *t*, respectively.

*3.3. Kriging Method*

The ordinary kriging interpolation method mainly involves performing interpolation calculations by searching for the spatial distance between the sites to be interpolated and known sites. This method not only quantifies the spatial autocorrelation between known points but also shows the spatial distribution of the target values [44]. The calculation equations are as follows:

$$Z(x_p) = \sum_{i=1}^{n} \lambda_i z(x_i) \tag{7}$$

In order to achieve unbiased estimations in kriging, the following set of equations should be solved simultaneously:

$$\sum_{i=1}^{n} \lambda_i \gamma(x_i, x_j) - \mu = \gamma(x_i, x_p) \tag{8}$$

where $j = 1, \ldots, n$ with $\sum_{i=1}^{n} \lambda_i = 1$. $Z(x_p)$ is the estimated value of variable $Z$ at location $x_p$; $z(x_i)$ is the known value at location $x_i$; $\lambda_i$ is the weight associated with the data; $\mu$ is the Lagrange coefficient; $\gamma(x_i, x_j)$ is the value of variogram corresponding to a vector with origin in $x_i$ and extremity in $x_j$; and n is the number of sampling points used in estimation.

## 4. Results and Discussion

*4.1. Evaluation of Multi-Source Satellite Data*

Based on the measured precipitation data obtained from the ground stations in the Lancang River Basin from 2016 to 2020, we evaluated various SPPs at daily, monthly, and annual scales (Figure 3). Considering that many "no rain" cases exist on a daily scale, the relative deviation calculation results have little significance; therefore, the CC and RMSE were used for the daily-scale evaluations, whereas the CC and RB were used for the monthly- and annual-scale evaluations.

In terms of the daily scale, FY-2G shows the lowest correlation with the ground-measured data and a high RMSE (Figure 3). Compared to that for the TRMM, the three types of GPM IMERG show significant improvements in CCs and error indices [45]. Notably, while the Early and Late Runs display similar performances, the Final Run demonstrates the strongest correlation with the measured data, indicating an ability to better reflect the observed precipitation characteristics. Nevertheless, the RMSE remains suboptimal. At the monthly scale, the CC of all SPPs is significantly higher than that at the daily scale, and the relative deviation is lower. FY-2G has the lowest CC, followed by TRMM 3B42RT. Both the GPM IMERG Early and Late Runs show the same CC values, whereas the Final Run has the highest CC of 0.87. Furthermore, the RB is greater than zero for all five SPPs, indicating an overestimation of ground-measured precipitation.

On an annual scale, the CC and RB are both lower than those on the monthly scale. The decline in CC can be attributed to the insufficient number of series used in the annual-scale assessment. As our study focused on newly available SPP data, we limited our selection to the past five years (since 2015), resulting in a relatively small dataset at the annual scale. Consequently, the SPPs may not accurately reflect the measured annual precipitation characteristics. For all SPPs, the RB value decreases gradually with an increase in temporal scale. TRMM 3B42RT and FY-2G show negative and low RB values, indicating that the measured precipitation is underestimated, whereas all three GPM types of IMERG have high positive RB values, which indicates that GPM IMERG can easily overestimate precipitation.

To further evaluate the impact of different precipitation levels on SPP performance, five indicators (POD, FAR, CSI, ETS, and FBI) were used to evaluate the threshold values of 0.1, 10, 25, and 50 mm/d, which delimit "rain/no rain", "light rain", "moderate rain", "heavy rain", and "torrential rain" conditions (Figure 4). All of the SPPs show the best detection ability for "rain/no rain" conditions, and the detection ability decreases with an increase in the rainfall level. The GPM Final Run shows the strongest detection ability, with the POD of "rain/no rain" conditions reaching 0.82. However, it tends to overestimate rainfall across different intensities based on the frequency deviation. The GPM Late and Early Runs show the second highest detection abilities. In their case, the "rain/no rain"

and "light rain" conditions are overestimated to a certain extent, whereas "moderate rain", "heavy rain", and "torrential rain" are underestimated. This indicates greater uncertainty in the detection of extreme precipitation and micro-precipitation using GPM products [46]. FY-2G shows the next-best detection capability. According to the FBI (FBI > 1: overestimate; FBI < 1: underestimate), in contrast to GPM IMERG, FY-2G tends to underestimate low-precipitation events in the form of drizzle, moderate rain, and heavy rain, but overestimates light rain and torrential rain. In addition, FY-2G commonly produces underreporting of precipitation events because the satellites typically use the bright temperature data of the infrared channel to invert precipitation while ignoring the influence of albedo information in the visible channel on precipitation, which results in a certain degree of detection error. Our results indicate that TRMM 3B42RT has low applicability at a high altitude and for complex terrains in the Lancang River Basin and can easily overestimate rainfall of a larger magnitude while underestimating rainfall of a smaller magnitude.

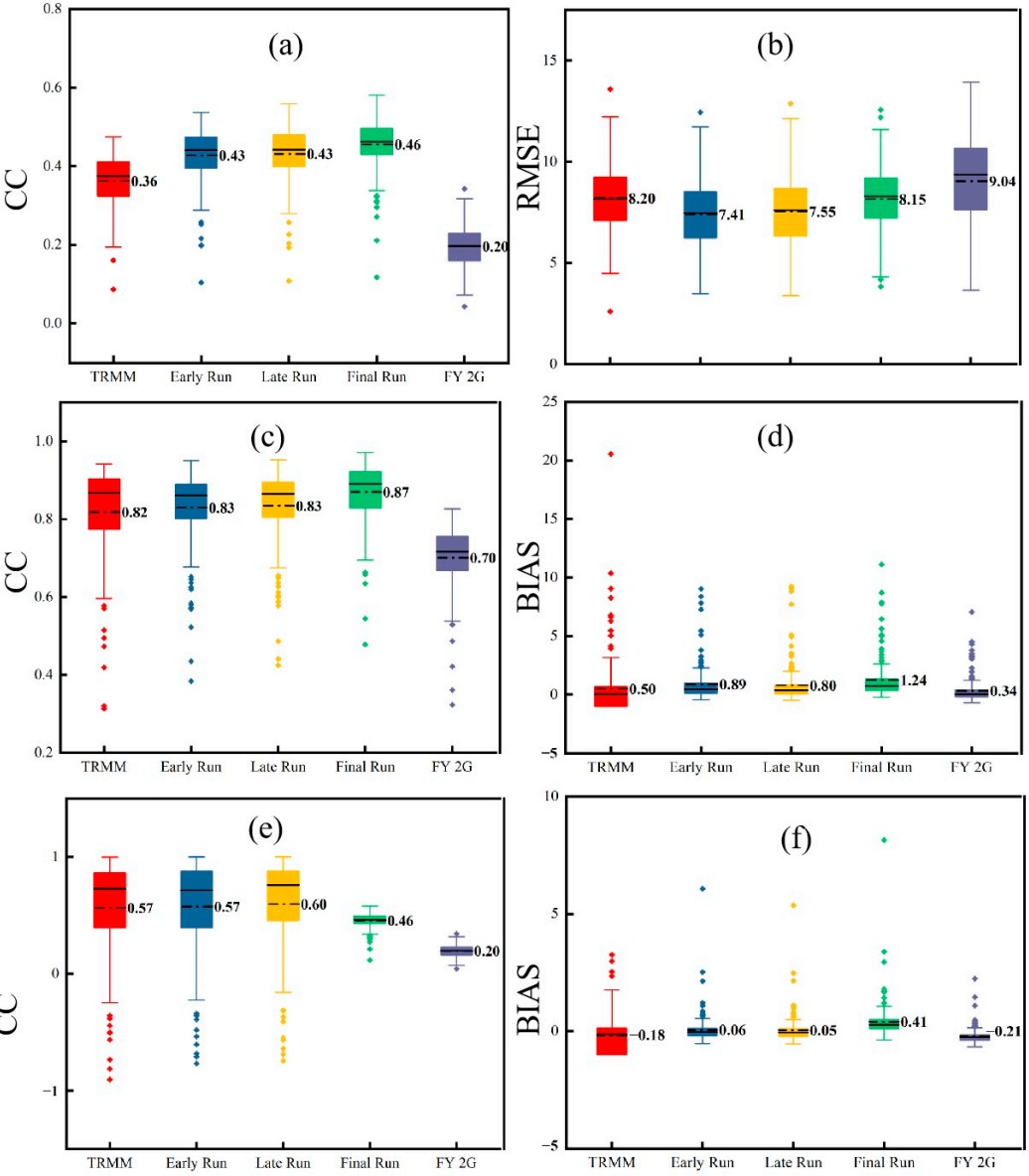

**Figure 3.** Box charts showing the evaluation results for five satellite precipitation products: (**a**) the results of CC at daily scale; (**b**) the results of RMSE at daily scale; (**c**) the results of CC at monthly scale; (**d**) the results of RB at monthly scale; (**e**) the results of CC at annual scale; (**f**) the results of RB at annual scale. The different boxes in the figure represent different satellite precipitation products. CC: correlation coefficient; RMSE: root mean square error; RB: relative bias.

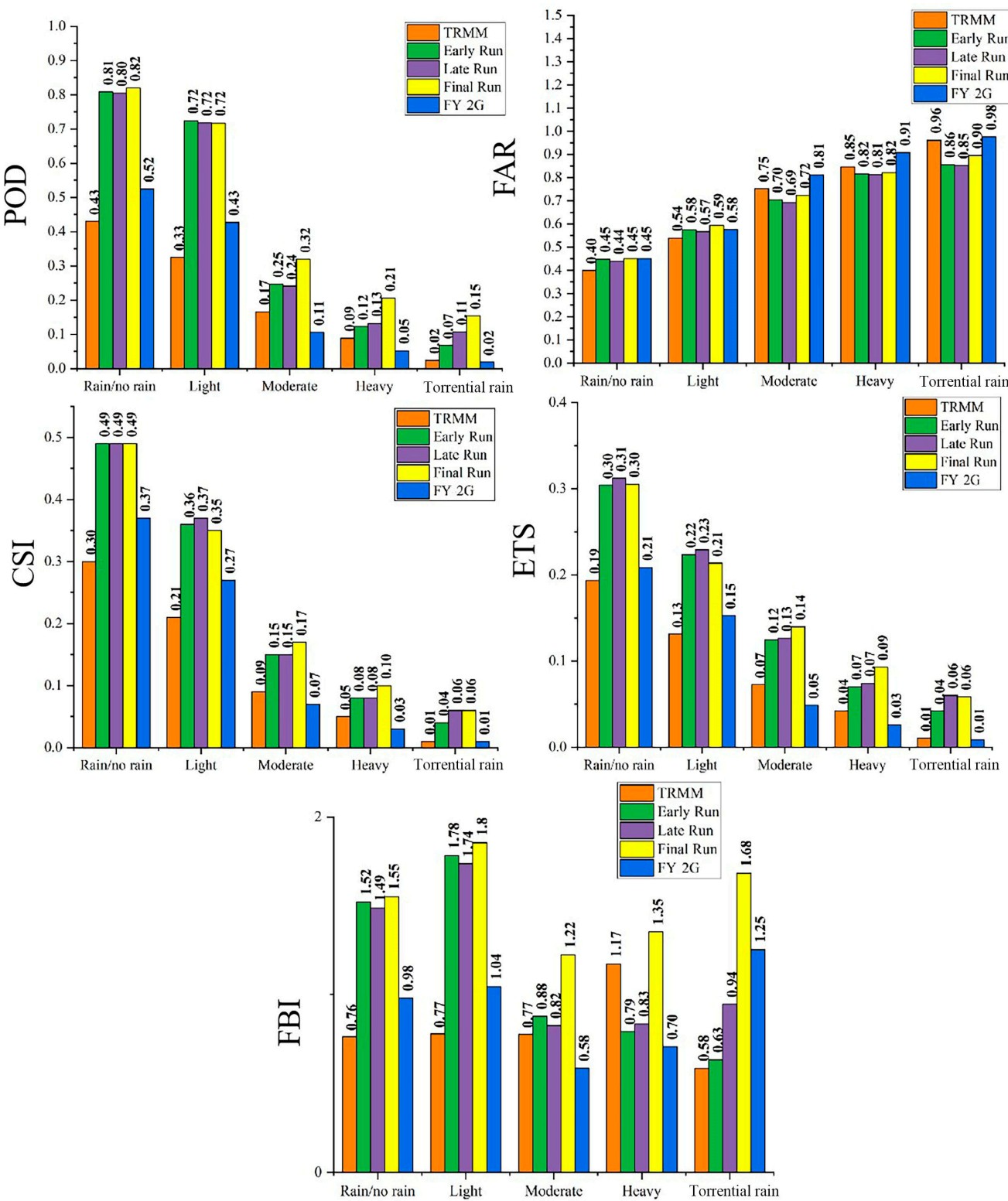

**Figure 4.** Radar map of SPP detection results for different precipitation grades. POD: probability of detection; FAR: false alarm rate; CSI: critical success index; ETS: fair precursor score; FBI: Frequency Bias Index.

### 4.2. Evaluation of the Bias Correction Scheme

After comprehensively considering the applicability of various SPPS to the Lancang River Basin, we selected the three types of GPM IMERG products and FY-2G for subsequent correction. We selected a period from 1 January 2020 to 31 December 2020 and a daily time scale. First, 176 ground observation stations in the Lancang River Basin were divided into a "correction group" and "verification group", with 88 stations in each group (Figure 5). Precipitation observations from the ground stations in the "correction group" were used to revise the satellite precipitation data, whereas the "verification group" was used to evaluate the correction effect of the window sliding method.

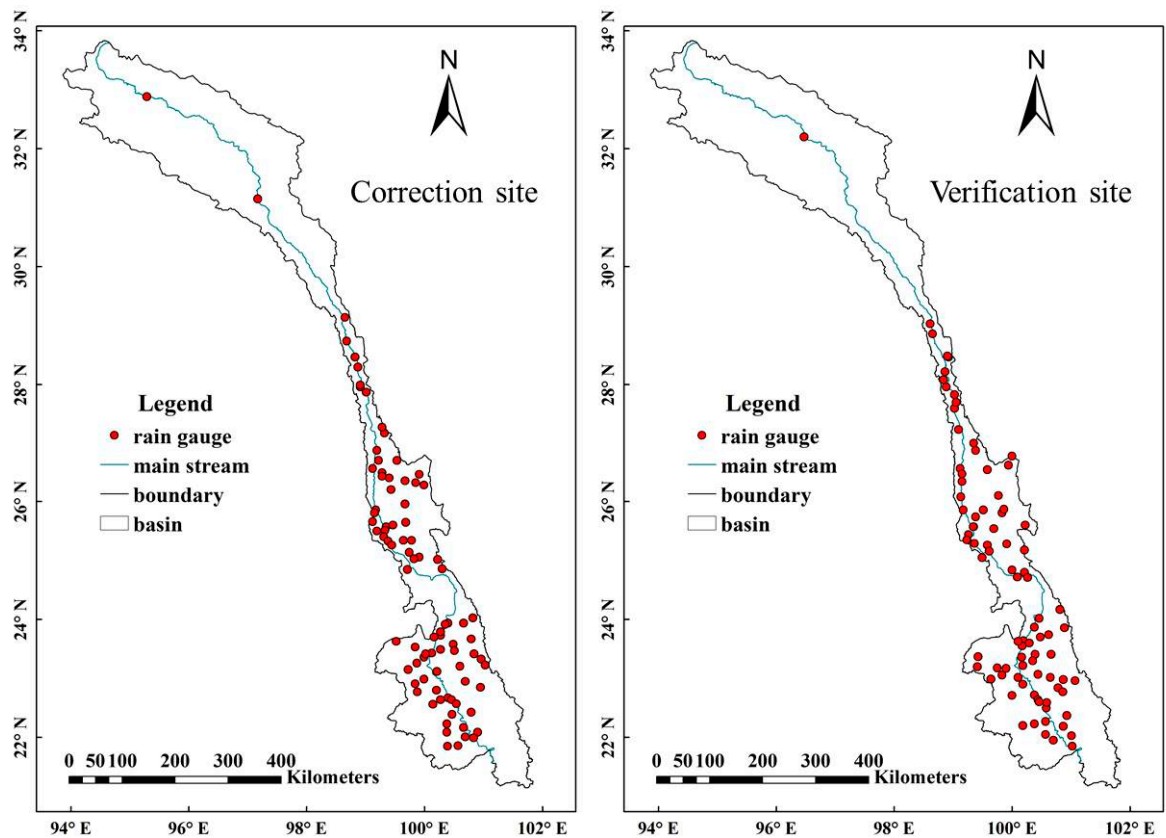

**Figure 5.** Map of correction and validation sites in the Lancang River Basin.

Second, we proposed a novel window sliding correction method that considers the spatial distribution of precipitation data. To improve the data correction effect, we determined the optimal window widths for the GPM and FY-2G products. The selection process is illustrated in Figure 6. For GPM IMERG, the optimal performance is achieved when the window side length is three grid units, yielding the highest CC and low error indexes. Conversely, for FY-2G, the best correction effect is attained when the side length of the correction window is 10 grid units, resulting in the highest CC and reduced RMSE and mean absolute error (MAE) values. Therefore, we selected three and ten grid units as the corrected window side lengths for GPM IMERG and FY-2G.

In this study, the CC, RMSE, POD, and MAE were used to evaluate the correction effects in various SPPs (Figure 7). In the Lancang River Basin, the proposed window sliding correction method improves the CC and precipitation detection ability (POD) and reduces the errors (RMSE and MAE) of the SPPs. It is worth mentioning that the correction method leads to the greatest CC improvement for FY-2G. GPM IMERG shows a large reduction in both error indicators, most obviously in the Final Run product. However, the four SPPs showed a low distinction in detection performance (POD) after the correction.

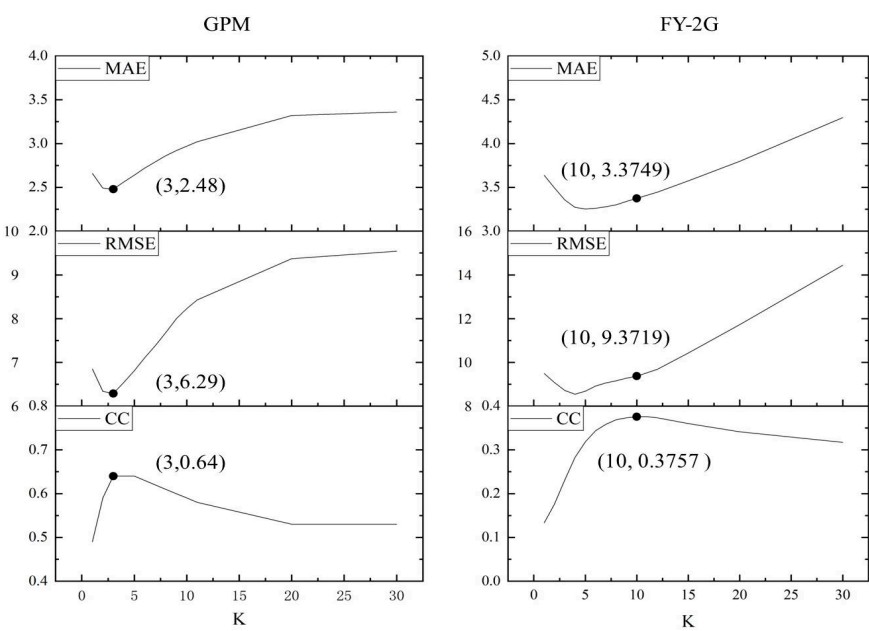

**Figure 6.** Correction effect of satellite precipitation products under different window side lengths.

**Figure 7.** Evaluation of the effect of correction for different satellite precipitation products.

### *4.3. Evaluation of the Precipitation Fusion Method*

To address different research needs, we adopted the Bayesian weighted average method to fuse the above deviation-corrected FY-2G and GPM products in different combinations. The optimal weight calculation results of the various fusion combinations are listed in Table 4. After fusing the daily-scale FY-Early, FY-Late, and FY-Final datasets based on the optimal weights, we calculated the CC, RMSE, MAE, and POD values to evaluate the effect of data fusion (note: the quality assessment of all kinds of datasets used 176 sites on the ground at one time without grouping) (Table 5). According to the results, the Bayesian fusion method further improves the precipitation data quality by increasing the CC and improving the detection performance of SPPs while also reducing the error between the SPP data and ground-measured data. However, the data quality is not significantly different between the fused products and the near real-time products of the GPM. Considering that the GPM IMERG Early Run is a near-real-time product with the lowest data delay, the FY-Early fusion dataset has more advantages in actual production if the timeliness of data acquisition is required.

**Table 4.** Optimal weight calculation results for fusion datasets.

| Datasets | FY 2G | GPM IMERG (Early/Late/Final) |
|---|---|---|
| FY-Early | 0.13 | 0.87 |
| FY-Late | 0.13 | 0.87 |
| FY-Final | 0.14 | 0.86 |

**Table 5.** Evaluation of the fusion effect for different satellite precipitation products.

| Datasets | CC | RMSE (mm) | MAE (mm) | POD |
|---|---|---|---|---|
| FY 2G corrected set | 0.40 | 9.49 | 3.36 | 0.37 |
| Early Run corrected set | 0.51 | 18.76 | 5.39 | 0.81 |
| Late Run corrected set | 0.53 | 20.77 | 5.63 | 0.82 |
| Final Run corrected set | 0.54 | 25.92 | 7.43 | 0.84 |
| FY-Early | 0.53 | 16.36 | 4.87 | 0.86 |
| FY-Late | 0.54 | 18.06 | 5.07 | 0.87 |
| FY-Final | 0.55 | 22.41 | 6.58 | 0.88 |

### *4.4. Spatial Effect of the Correction–Fusion Method*

To further explore the effect of the correction–fusion method at the catchment scale, we analyzed spatial changes in the GPM IMERG Early Run, Late Run, Final Run, and FY-2G datasets before and after correction using ordinary kriging interpolation. The results are shown in Figure 8. Early, Late, and Final represent the evaluation results of the original SPPs, and C-Early, C-Late, and C-Final represent the evaluation results of the revised SPPs. The correction effects of GPM and FY-2G are spatially consistent and more obvious in the middle and lower reaches of the Lancang River Basin than in the upper reaches. This is related to the landforms and distribution of ground stations in the Lancang River Basin. First, the upper reaches are high altitudes with a complex topography, and the quality of ground-measured data and SPPs in this region is low; therefore, it is difficult to obtain an efficient correction. In contrast, the terrain of the middle and lower reaches is less complex; thus, the accuracy of the ground and satellite data is higher, which is conducive to the correction of local SPPs. Second, the essence of the window sliding correction method is to perform reasonable corrections of SPPs based on measured precipitation data obtained from ground stations. However, few ground observation stations exist in the upper reaches of the Lancang River Basin, which causes significant difficulties in satellite data correction. Thus, the advantages of the window sliding correction method are not reflected in this region. In contrast, the middle and lower reaches of the Lancang River Basin have more ground observation stations with a denser distribution. This provides richer ground reference information for data correction, reducing the difficulty of data correction and improving

the effect of data correction. Thus, although the proposed method can significantly improve data quality, the degree of improvement depends on the number of ground sites [47]. However, the correction method used by Lu et al. is applicable to monthly or longer time scales, whereas the method proposed in this study is more applicable to daily-scale dataset correction, which represents an improvement from previous studies.

To further improve the data quality, we performed fusion based on the previous correction procedure. Figure 9 depicts the spatial distribution characteristics of the CC for the datasets before and after fusion (in which, for example, C-FY-2G represents the evaluation result of FY-2G after the correction, C-Early represents the evaluation result of the Early Run after the correction, and FY-Early represents the evaluation results of the revised FY-2G and Early Run fusion dataset). The Bayesian fusion method improves data quality and generates a more precise dataset. Similar to the conclusions of Wanders et al. [48], data fusion can reduce precipitation uncertainty. In addition, the fusion datasets obtained in this study exhibit high CC values in the middle and lower reaches of the Lancang River, indicating a better ability to reflect the precipitation characteristics of the region, but with less ideal CC values in the upper reaches. This is attributed to the relatively complex precipitation, influenced by climate, terrain, and other factors, the low accuracy of the original data in areas with few ground observation stations, and the relative lack of ground-measured reference data. Thus, the applicability of the Bayesian fusion method in this region requires further study.

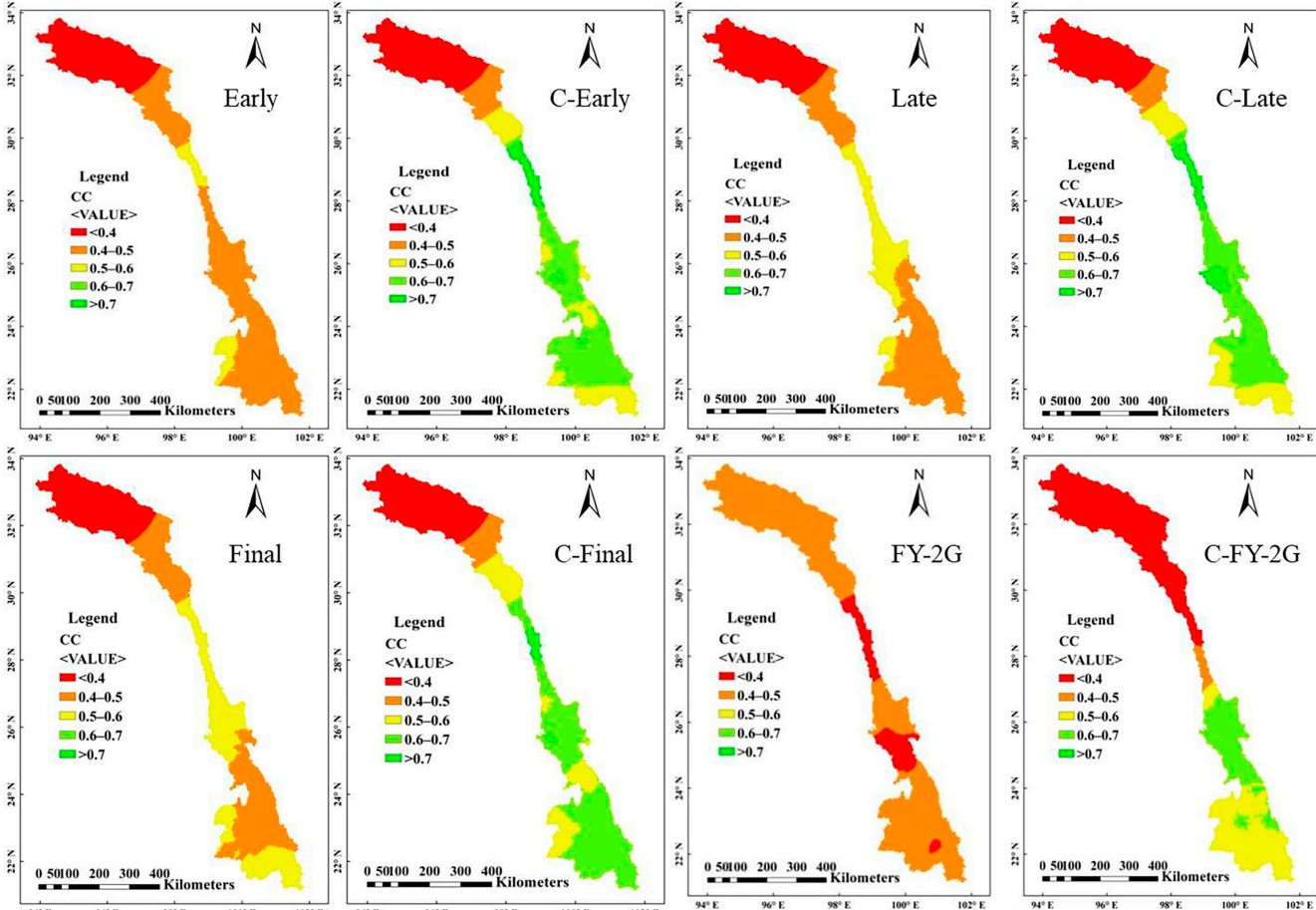

**Figure 8.** Spatial distribution of the correction effects for different satellite precipitation products.

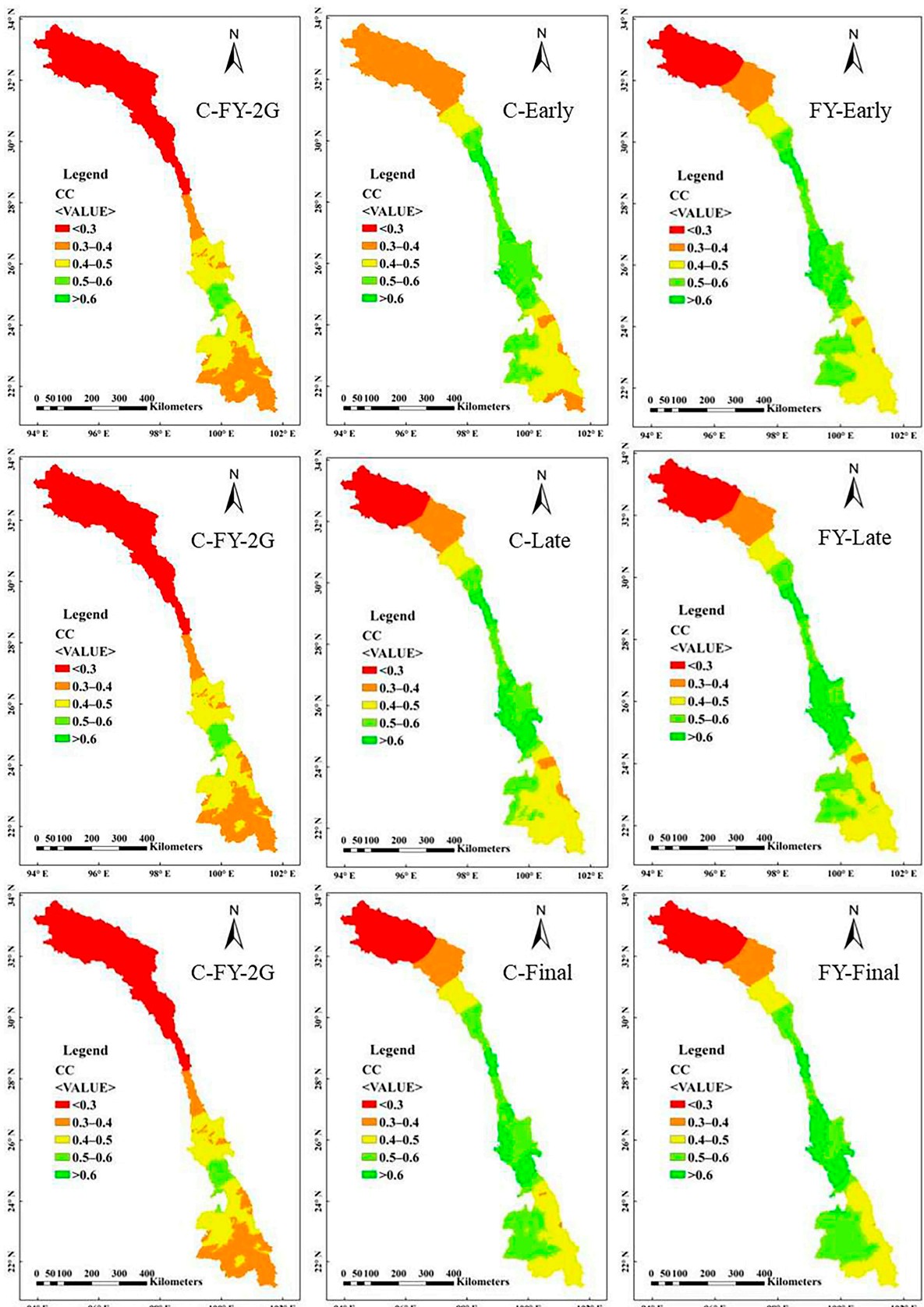

**Figure 9.** Spatial distribution of CC values before and after fusion for different satellite precipitation products.

In summary, the correction–fusion method proposed in this study can reduce the systematic errors of SPPS and provide alternative data sources for ground observation stations, which is an optimization of the findings of Zhang et al. [49]. However, like the method used by Xiao et al. [50], the proposed correction–fusion method also depends on the density of ground stations; hence, it is not suitable for use in areas lacking ground observation stations. Additionally, the validity of the correction method depends on the spatial consistency of the deviation [51].

## 5. Future Research Directions

In recent years, satellite remote sensing technology has been improving in terms of its spatiotemporal resolution, high levels of precision, and dynamic capabilities, which have greatly improved our capability to observe Earth. SPPs are widely used for precipitation forecasting, water resource management, and flood disaster monitoring. In this study, we evaluated the accuracy and applicability of multi-source satellite precipitation data to the Lancang River Basin, then performed correction and fusion procedures on the optimal satellite data based on ground observation precipitation data, thereby expanding the local data sources. However, this study has the following limitations, which suggest the need for further studies:

(1) Spatial scale: the spatial scale of the satellite precipitation data used in this study was $0.1°$ (approximately 10 km); therefore, its application on a smaller spatial scale (such as 5 km) requires further verification and analysis.

(2) Time scale: satellite precipitation data with a time scale of 1 day were selected for the comparative evaluation and applicability analysis in this study, which cannot easily meet the needs of flood monitoring and forecasting. In addition, the effect of the deviation correction is closely related to the time scale [52]. The next step is to research on an hourly time scale to better meet the demands of flood monitoring, forecasting, and management.

(3) Precipitation level: Deng et al. [53] found that some precipitation events were lost due to systematic errors in the revised model. Therefore, follow-up research should determine whether the proposed correction–fusion processing improves the ability of SPPs to detect precipitation events.

(4) Combining multiple factors: In this study, we aimed to preliminarily verify the improvement effect of the proposed correction–fusion method on SPPs. Therefore, according to actual data from the Lancang River Basin, a correction was performed only through the relationship between ground station data and multi-source satellite precipitation data. However, satellites can only detect local precipitation conditions over a period; that is, they reflect a transient situation [54]. Therefore, the use of the relationship between satellite precipitation data and ground-measured data for corrections is limited. Future research should refer to the work of Zhang et al. [55] and other studies and attempt to add multiple factors to further improve data quality, such as latitude and longitude, digital elevation model topographic factors, seasonal factors, and the normalized difference vegetation index.

(5) Change study area: The method of this study will be further improved and applied to other remote areas with insufficient data to verify the universality and reliability of the method on the one hand and to provide precipitation data sources for other similar areas on the other hand.

## 6. Conclusions

To ensure the complementary performance of domestic (FY-2G) and foreign SPPs in China, we evaluated the accuracy and applicability of precipitation data from the TRMM 3B42RT, GPM IMERG Early Run, GPM IMERG Late Run, GPM IMERG Final Run, and FY-2G satellites, then developed a novel correction–fusion method to improve the data quality. The main conclusions are as follows:

(1) The correlation between the SPP data and ground-measured data in the Lancang River Basin was higher for the GPM IMERG products than for the TRMM and FY-2G, indicating a better ability to reflect actual long-term precipitation characteristics. Among the GPM products, the Final Run showed a better performance than the Early and Late Runs, which exhibited similar performances. However, FY-2G exhibited a lower RB among the SPPs on monthly and annual scales, which indicates that FY-2G products can better describe total precipitation.

(2) We proposed a novel window sliding data correction method that significantly improved the quality of SPP data by not only improving their correlation and detection ability but also reducing their deviation. This method showed some applicability to the Lancang River Basin, although the correction effect was better in the middle and lower reaches of the basin than in the upper reaches due to the higher number of ground observation stations and higher quality of the ground reference information.

(3) The Bayesian fusion method further improved the data quality and provided more reliable data sources for the Lancang River Basin. In this study, the corrected FY-2G data were fused with selected GPM-revised products for the first time to obtain near- and non-real-time fusion datasets. The former datasets are suitable for scenarios requiring more timely acquisition, such as practical applications, whereas the latter are more useful for scenarios prioritizing accuracy over data timeliness, such as scientific research.

**Author Contributions:** Conceptualization, M.Y. and H.W. (Hao Wang); Methodology, H.W. (Hao Wang); Validation, L.N.; Resources, H.W. (Hejia Wang); Data curation, H.W. (Hejia Wang); Writing—original draft, L.N.; Writing—review & editing, L.N.; Supervision, M.Y. and N.D.; Funding acquisition, H.W. (Hejia Wang). All authors have read and agreed to the published version of the manuscript.

**Funding:** This work was supported by the Water Science and Technology Project of Ordos City (No:ESKJ2023-001); the Science and Technology Project of China Huaneng Group Research on Integrated Meteorology and Hydrology Forecasting System in Lancang River Basin (No:HNKJ21-HF241); China Power Construction Corporation Technology Project (DJ-HXGG-2021-04) and Key R&D Plan Project in Yunnan Province (202203AA080010).

**Data Availability Statement:** The original contributions presented in the study are included in the article, further inquiries can be directed to the corresponding author.

**Conflicts of Interest:** The authors declare that they have no known competing financial interests or personal relationships that could have appeared to influence the work reported in this paper.

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
