# Peer review of "An Innovative Correction–Fusion Approach for Multi-Satellite Precipitation Products Conditioned by Gauge Background Fields over the Lancang River Basin"

_remotesensing, doi:10.3390/rs16111824_

Round 1

Reviewer 1 Report

Comments and Suggestions for Authors

This study evaluated different SPPs with ground observations and proposed the window sliding data correction method to correct the SPPs. With the correction and Bayesian data fusion scheme, it provided more reliable data for the Lancang River Basin.

1. Line 143,151 What are the strict quality control for SPPs? Did you operate them? How about ground station data QC?

2. Line 169-170, add the ETS. Should the "Hc" in ETS equation be "Hs"?

3. How to determine the initial wind width? in kilometers or  grid units? What is the assessment requirement?

4. There are two plots in Figure2, no legend for the left one. There is no caption about m, n in Eq. 4. How to correct the raster data in the window which window has no ground observation? no correction?

5.  Please uniform the variable writing format in Table 3, such as g.

6. Line 273, I didn't see the equation 5,6 from reference 42. If it isn't  proposed by you, please mark the exact reference here.

7. What is the meaning of P? Is it a constant or something else?  What is the Si? I think Z is not the subscript of P in Eq. 5.

8. Line 282, the evaluation period is 2016 to 2020, but the time of TRMM data is only 2016 to 2019 shown in Table 1. The samples number are different.

9.Figure 3 is not clear enough, the colors are light. How about the RMSE of monthly and annual evaluations?

10. Figure 4 is hard to read. The conclusion is difficult to understand from the figure. Different colors for different SPPs overlay together, and no panel label a, b, c, d. Please change Figure 4 to histogram plot or other type, and add the figure reference in description sentences, e.g. xxxx in Figure 4(a).

11. Line 321, the "frequency deviation"  is CSI?

12.Most figures don't have panel label and the legend is not clear, please add the label, the variable units and modify the legends, such as Figure 1,2,4,5,6,7,8,9.

13. Line 354, 357, "three grid units"->"3 grid units", which is consistent writing format with 10 grid units.

14. Figure 6 shows 3 indexes, but it looks like only the highest CC is used as the final index for the window side length choice? Does the left plot in Figure 6 show the average statistic for 3 GPM SPPs ?

15. Line 361, RMSE? POD?

16. The "revised set"  in Table 5 is the corrected set, right? But the results of indexes are different from Figure 7, why?

17. Why do you fuse FY-2G and each GPM SPPs, not two or three GPM SPPs?

18.Figure 8 shows the worse result after correction than before correction for FY-2G in the upper reaches. Why? If no correction will be made in no ground data window, the CC should be the same as before, not worse than before.

19. Section 4.5 is the discussion, please move this section to section 5.

20. Line 487, "FY-2G exhibited the smallest errors among the four SPPs", I didn't see this conclusion in text before. 

21. There are many similar sentences, just like "consistent with the findings/ conclusions/result of xxx", such as line 290, 324, 413, 425, 436. It looks like all the conclusions have been verified before. Please modify or delete these sentences. 

Author Response

Thank you for your comments concerning our manuscript “An innovative correction–fusion approach for multi-satellite precipitation products conditioned by gauge background fields over the Lancang River Basin”, that we submitted to remotesensing (remotesensing-2972194).

We found the reviewers’ comments to be very valuable and helpful in improving our presentation, as well as important for guiding significantly to our research. We have read the comments carefully and the manuscript have been thoroughly rechecked and provided in the revised manuscript (red highlighted) according to these comments and suggestion. Attached please find our revised manuscript, and listed below are our point-by-point responses to the reviewer suggestions, which are highlighted in red. Thank you again for handling this manuscript.

With all best wishes, 

                         Yours sincerely

Linjiang Nan

Point 1: Line 143,151 What are the strict quality control for SPPs? Did you operate them? How about ground station data QC?

Response 1: Data quality control is an important part of data management, which refers to the monitoring and improvement of data sources to ensure the accuracy, integrity, consistency and reliability of data.

All kinds of data sources used in this paper have been quality controlled strictly by the provider (such as National meteorological department), so I didn’t reoperate them.

The ground observation data come from the National Meteorological Information Center and Lancang River Group, and have passed the data quality control, so as to ensure the continuity of the data in time and the reliability in accuracy.

Point 2: Line 169-170, add the ETS. Should the "Hc" in ETS equation be "Hs"?

Response 2: I have revised this part according to your valuable advice in the manuscript, which are highlighted in red. (The new manuscript appears in lines 234-235).

Point 3: How to determine the initial wind width? in kilometers or  grid units? What is the assessment requirement?

Response 3: The initial window width refers to a grid cell (belonging to grid units).

Correction and effect evaluation will be carried out immediately after each window size adjustment. The effect is shown in Figure 6, and the window with the highest correlation coefficient and the lowest error is finally selected.

Point 4: There are two plots in Figure2, no legend for the left one. There is no caption about m, n in Eq. 4. How to correct the raster data in the window which window has no ground observation? no correction?

Response 4: I have revised this part according to your valuable advice in the manuscript.

Figure 2 is a simple diagram. In the actual correction process of this study, the correction window size was constantly adjusted so that at least one ground observation station was included in the window to correct the satellite data.

Point 5: Please uniform the variable writing format in Table 3, such as g.

Response 5: I have revised this part according to your valuable advice in the manuscript, which are highlighted in red. (The new manuscript appears in lines 323-353).

Point 6: Line 273, I didn't see the equation 5,6 from reference 42. If it isn't  proposed by you, please mark the exact reference here.

Response 6: I have revised this part according to your valuable advice in the manuscript, which are highlighted in red. (The new manuscript appears in lines 361-371 and 823-825).

Point 7: What is the meaning of P? Is it a constant or something else?  What is the Si? I think Z is not the subscript of P in Eq. 5.

Response 7: P is the data fusion result (new data set), not a constant, D is the measured rainfall data (origin data set). In order to distinguish it from precipitation, P here is modified to B. I have revised this part according to your valuable advice in the manuscript, which are highlighted in red. (The new manuscript appears in lines 304-334).

The Si and Z you mentioned are not found in Eq. 5.

Point 8: Line 282, the evaluation period is 2016 to 2020, but the time of TRMM data is only 2016 to 2019 shown in Table 1. The samples number are different.

Response 8: Because the TRMM satellite was decommissioned at the end of 2019, the TRMM 3B42RT dataset was only updated until December 31, 2019.

Point 9: Figure 3 is not clear enough, the colors are light. How about the RMSE of monthly and annual evaluations?

Response 9:

Considering that many “no rain” cases exist on a daily scale, the relative deviation calculation results have little significance; therefore, CC and RMSE were used for daily-scale evaluations, whereas CC and RB were used for monthly- and annual-scale evaluations.

Point 10: Figure 4 is hard to read. The conclusion is difficult to understand from the figure. Different colors for different SPPs overlay together, and no panel label a, b, c, d. Please change Figure 4 to histogram plot or other type, and add the figure reference in description sentences, e.g. xxxx in Figure 4(a).

Response 10: I have revised this part according to your valuable advice in the manuscript.

Point 11: Line 321, the "frequency deviation"  is CSI?

Response 11: the "frequency deviation"  refers to the evaluation index FD, the chart has been added in the corresponding position.

Point 12: Most figures don't have panel label and the legend is not clear, please add the label, the variable units and modify the legends, such as Figure 1,2,4,5,6,7,8,9.

Response 12: I have revised this part according to your valuable advice in the manuscript.

Point 13: Line 354, 357, "three grid units"->"3 grid units", which is consistent writing format with 10 grid units.

Response 13: I have revised this part according to your valuable advice in the manuscript, which are highlighted in red. (The new manuscript appears in line 463).

Point 14: Figure 6 shows 3 indexes, but it looks like only the highest CC is used as the final index for the window side length choice? Does the left plot in Figure 6 show the average statistic for 3 GPM SPPs ?

Response 14: Figure 6 shows the MAE, RMSE and CC results of GPM products and FY-2G under different window widths. The selection is based on the principle of the highest CC and the lower MAE and RMSE, and the highest CC is taken as the first selection criterion. Because CC can reflect the closeness of the correlation between satellite products and ground measured data, MAE and RMSE have no obvious changes when the window width is less than 10 in this study, so CC is preferred to be compared.

Yes. Because the data structures of the three categories of GPM products (early, late, and final) are consistent, the correction window for GPM products can be determined by averaging them at first.

Point 15: Line 361, RMSE? POD?

Response 15: I have revised this part according to your valuable advice in the manuscript, which are highlighted in red. (The new manuscript appears in lines 472-480).

Point 16: The "revised set"  in Table 5 is the corrected set, right? But the results of indexes are different from Figure 7, why?

Response 16: Yes. I have revised this part according to your valuable advice in the manuscript, which are highlighted in red.

Because figure 7 is the part of the window slide correction, the effect evaluation of this part is as follows (as shown in lines 449-455 of the new manuscript): First, 176 ground observation stations in the Lancang River Basin were divided into a "correction group" and "verification group,” with 88 stations in each group (Figure 5). Precipitation observations from ground stations in the "correction group" were used to revise the satellite precipitation data, whereas the "verification group" was used to evaluate the correction effect of the window sliding method. But table 5 is the part of Bayesian fusion, and the quality assessment of all kinds of data sets used 176 sites on the ground at one time, without grouping. (Elaboration has been added to lines 491-492 of the new manuscript).

Point 17: Why do you fuse FY-2G and each GPM SPPs, not two or three GPM SPPs?

Response 17: Because the three types of GPM have their own characteristics through our evaluation, this paper intends to absorb the advantages of these three types of foreign products one by one based on domestic FY-2G products, so as to obtain near-real-time and non-real-time fusion data sets to solve different practical problems. Of course, your question is very valuable. In the future research, we will fully explore the advantages of multiple precipitation products according to the data acquisition situation, so as to realize the integration of multi-source data (larger than two datasets) and further obtain new and reliable precipitation data sources.

Point 18: Figure 8 shows the worse result after correction than before correction for FY-2G in the upper reaches. Why? If no correction will be made in no ground data window, the CC should be the same as before, not worse than before.

Response 18: Because this is related to the landforms and distribution of ground stations in the Lancang River Basin. First, the upper reaches are high altitudes with complex topography, and the quality of ground-measured data and SPPs in this region is low; therefore, it is difficult to obtain an efficient correction. Second, the essence of the window sliding correction method is to perform reasonable correction of SPPs based on measured precipitation data obtained from ground stations. However, few ground observation stations exist in the upper reaches of the Lancang River Basin, which causes significant difficulties in satellite data correction. Thus, the advantages of the window sliding correction method are not reflected in this region.

Because the window widths of GPM and FY-2G were selected respectively (as shown in 461-468 of the new manuscript) before the window sliding correction was carried out in this paper, it can be seen that for GPM and FY-2G, the width of the correction window is different. GPM is 3, FY-2G is 10, and the window width of FY-2G is larger, which can cover more ground observation stations. So there are changes before and after correction. However, there are few ground observation stations in the upstream area, the correction effect in this area is not ideal.

Point 19: Section 4.5 is the discussion, please move this section to section 5.

Response 19: I have revised this part according to your valuable advice in the manuscript, which are highlighted in red. (The new manuscript appears in lines 578 and 619).

Point 20: Line 487, "FY-2G exhibited the smallest errors among the four SPPs", I didn't see this conclusion in text before.

Response 20: Conclusion (1) corresponds to article 4.1 Evaluation of multi-source satellite data. Figure 3 shows that for both monthly and annual scales, RB of FY-2G is low, so this conclusion is reached. In combination with your valuable questions and in order to express with more rigorousness, the conclusions here have been improved, as shown in lines 631-633 of the new manuscript.

Point 21: There are many similar sentences, just like "consistent with the findings/ conclusions/result of xxx", such as line 290, 324, 413, 425, 436. It looks like all the conclusions have been verified before. Please modify or delete these sentences.

Response 21: I have revised these parts according to your valuable advice in the manuscript, which are highlighted in red. (The new manuscript appears in lines 384, 426, 541, 556, 569).

Reviewer 2 Report

Comments and Suggestions for Authors

[Title]

An innovative correction–fusion approach for multi-satellite precipitation products conditioned by gauge background fields over the Lancang River Basin

[Summary]

The author evaluated different SPPs (TRMM/GPM/FY-2G) over the Lancang River Basin. Then, they prposeed a novel correction–fusion method based on window sliding data correction and Bayesian data fusion. The fusion process of GPM and FY-2G improve the correlation between ground observation and SPP data.

[Major Comment]

The sturucture of the presentation is fine, and I have two major comment.

First. The reason(s) why you focus on the Lancang River Basin as the study area is/are not clear although the remoteness and the lack of observation in the area is mentioned in the introduction. The reason (altough mentined in subsection 2.1) should be stated in the introduction in order to help all readers, including people not knowing the geophysical characteristics of the area well, understand the scientific significance of studying the area.

Second. You mention the future research directions in subsection 4.5. The main stream is improve of the correction and fusion method, but do you intend to use further SPP data in time and areas? Corresponding to the 1st comment, there should be remote and sparse-gauge area other than the Lancang River Basin. If you consider to use further data of other areas, it may be better to discuss in the subsection 4.5.

[Minor Coments]

Line 33, 39, 67 and others: (cite)

   Do you intend to cite previous studies?

Figure 4

   It is hard to distinguish SPPs because the colors are overlaid and mixed. I reccomend to use not radar chart but bar chart.

Comments on the Quality of English Language

[English Error]

English is almost fine, but there are some tiny mistakes.

Table 2

   The "Hs" of ETS is Hc?

Line 321

   GMP -> GPM

Author Response

Thank you for your comments concerning our manuscript “An innovative correction–fusion approach for multi-satellite precipitation products conditioned by gauge background fields over the Lancang River Basin”, that we submitted to remotesensing (remotesensing-2972194).

We have revised the manuscript, and would like to re-submit it for your consideration. We have addressed the comments raised by the reviewers, and the amendments are highlighted in red in the revised manuscript. Point by point responses to the reviewers’ comments are listed below this letter. We hope that the revised version of the manuscript is now acceptable for publication in your journal.

I look forward to hearing from you soon.

With best wishes,

Yours sincerely

Linjiang Nan

Major:

Point 1: The reason(s) why you focus on the Lancang River Basin as the study area is/are not clear although the remoteness and the lack of observation in the area is mentioned in the introduction. The reason (altough mentined in subsection 2.1) should be stated in the introduction in order to help all readers, including people not knowing the geophysical characteristics of the area well, understand the scientific significance of studying the area.

Response 1: I have revised this part according to your valuable advice in the manuscript, which are highlighted in red. (The new manuscript appears in lines 104-119).

Point 2: You mention the future research directions in subsection 4.5. The main stream is improve of the correction and fusion method, but do you intend to use further SPP data in time and areas? Corresponding to the 1st comment, there should be remote and sparse-gauge area other than the Lancang River Basin. If you consider to use further data of other areas, it may be better to discuss in the subsection 4.5.

Response 2: We intend to use further SPP data in time and areas, and I have revised this part according to your valuable advice in the manuscript, which are highlighted in red. (The new manuscript appears in lines 629-632).

Minor:

Point 3: Line 33, 39, 67 and others: (cite)

Do you intend to cite previous studies?

Response 3: I have revised this part according to your valuable advice in the manuscript, which are highlighted in red.

Point 4: Figure 4

It is hard to distinguish SPPs because the colors are overlaid and mixed. I reccomend to use not radar chart but bar chart.

Response 4: I have revised this part according to your valuable advice in the manuscript.

Point 5: English is almost fine, but there are some tiny mistakes.

Table 2

   The "Hs" of ETS is Hc?

Line 321

   GMP -> GPM

Response 5: I have revised this part according to your valuable advice in the manuscript, which are highlighted in red. (The new manuscript appears in line 435).

Reviewer 3 Report

Comments and Suggestions for Authors

This research introduces a new fusion-correction method for satellite precipitation products (SPP) that significantly improves the accuracy and reliability of precipitation data in the Lancang River Basin, a region with sparse ground observations.

Unlike previous studies, this approach combines window slip data correction with Bayesian data fusion tailored to the unique geographical and climatic conditions of the area. By effectively merging corrected datasets from different satellites, including GPM IMERG and FY-2G, the study provides a more accurate and reliable source of precipitation data, addressing a critical need for improved water resource management and forecasting in data deficient regions.

This method stands out by leveraging the strengths of multiple satellite datasets and ground measurements, providing a significant advance over existing single-source or less sophisticated fusion techniques.

I appreciate that the authors highlight the importance of using ground-level monitoring.

Revisions:

- The figures need to be redone to a much better resolution;

- Figure 3 should be enriched with information. For example, authors must display on chart boxes maximums, average values and what they consider important. However, for me at least it is not clear from figure 3 and the paper, what the correlation coefficient represents and how exactly it was calculated because in Table 2, range appears, from -1 to 1. Same observations for figure 7. 

Author Response

Thank you for your comments concerning our manuscript “An innovative correction–fusion approach for multi-satellite precipitation products conditioned by gauge background fields over the Lancang River Basin”, that we submitted to remotesensing (remotesensing-2972194).

We found the reviewers’ comments to be very valuable and helpful in improving our presentation, as well as important for guiding significantly to our research. We have read the comments carefully and the manuscript have been thoroughly rechecked and provided in the revised manuscript (red highlighted) according to these comments and suggestion. Attached please find our revised manuscript, and listed below are our point-by-point responses to the reviewer suggestions, which are highlighted in red. Thank you again for handling this manuscript.

With all best wishes, 

                         Yours sincerely

Linjiang Nan

Point 1: The figures need to be redone to a much better resolution;

Response 1: I have redone the figures according to your valuable advice in the manuscript.

Point 2: Figure 3 should be enriched with information. For example, authors must display on chart boxes maximums, average values and what they consider important. However, for me at least it is not clear from figure 3 and the paper, what the correlation coefficient represents and how exactly it was calculated because in Table 2, range appears, from -1 to 1. Same observations for figure 7.

Response 2: I have optimized the figures according to your valuable advice in the manuscript.
